# Robot Learning in Homes:
# Improving Generalization and Reducing Dataset Bias

**Abhinav Gupta**[*]     **Adithyavairavan Murali**[*]     **Dhiraj Gandhi**[*]     **Lerrel Pinto**[*]

The Robotics Institute
Carnegie Mellon University

## Abstract

Data-driven approaches to solving robotic tasks have gained a lot of traction in recent years. However, most existing policies are trained on large-scale datasets collected in curated lab settings. If we aim to deploy these models in unstructured visual environments like people's homes, they will be unable to cope with the mismatch in data distribution. In such light, we present the first systematic effort in collecting a large dataset for robotic grasping in homes. First, to scale and parallelize data collection, we built a low cost mobile manipulator assembled for under $3K$ USD. Second, data collected using low cost robots suffer from noisy labels due to imperfect execution and calibration errors. To handle this, we develop a framework which factors out the noise as a latent variable. Our model is trained on $28K$ grasps collected in several houses under an array of different environmental conditions. We evaluate our models by physically executing grasps on a collection of novel objects in multiple unseen homes. The models trained with our home dataset showed a marked improvement of 43.7% over a baseline model trained with data collected in lab. Our architecture which explicitly models the latent noise in the dataset also performed 10% better than one that did not factor out the noise. We hope this effort inspires the robotics community to look outside the lab and embrace learning based approaches to handle inaccurate cheap robots.

## 1 Introduction

Powered by the availability of cheaper robots, robust simulators and greater processing speeds, the last decade has witnessed the rise of data-driven approaches in robotics. Instead of using hand-designed models, these approaches focus on the collection of large-scale datasets to learn policies that map from high-dimensional observations to actions. Current data-driven approaches mostly focus on using simulators since it is considerably less expensive to collect simulated data than on an actual robot in real-time. The hope is that these approaches will either be robust enough to domain shifts or that the models can be adapted using a small amount of real world data via transfer learning. However, beyond simple robotic picking tasks [1, 2, 3], there exist little support to this level of optimism. One major reason for this is the wide "reality gap" between simulators and the real world.

Therefore, there has concurrently been a push in the robotics community to collect real-world physical interaction data [4, 5, 6, 7, 8, 9, 10, 11] in multiple robotics labs. A major driving force behind this effort is the declining costs of hardware which allows scaling up data collection efforts for a variety of robotic tasks. This approach has indeed been quite successful at tasks such as grasping, pushing, poking and imitation learning. However, these learned models have often been shown to overfit (even after increasing the number of datapoints) and the performance of these robot learning methods tends

---

[*]Equal contribution. Direct correspondence to: `{abhinavg,amurali,dgandhi,lerrelp}@cs.cmu.edu`

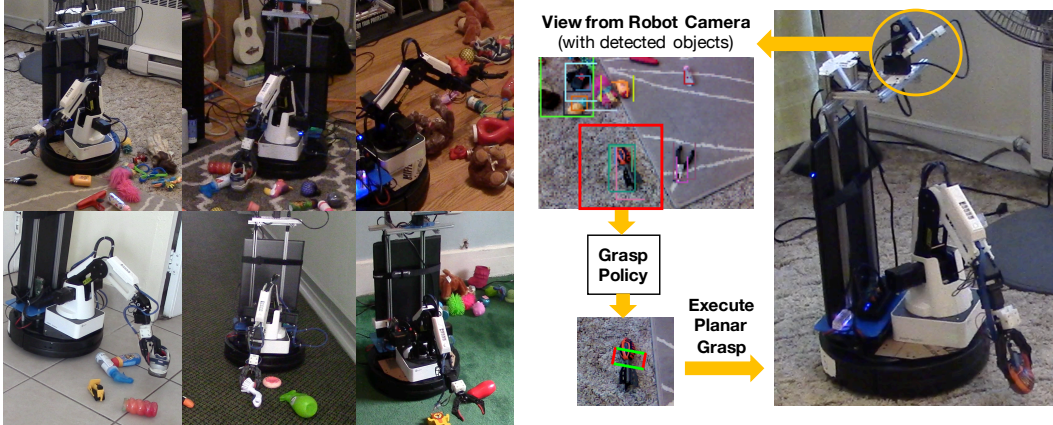

Figure 1: We built multiple low-cost robots and collected a large grasp dataset in several homes.

to plateau fast. This leads us to an important question: why does robotic action data not lead to similar gains as we see in other prominent areas such as computer vision [12] and natural language processing [13]?

The key to answering this question lies in the word: "real". Many approaches claim that the data collected in the lab is real-world data. But is this really true? How often do we see white table-clothes or green backgrounds in real-world scenarios? In this paper, we argue that current robotic datasets lack the diversity of environments required for data-driven approaches to learn invariances. Therefore, the key lies in moving data collection efforts from a lab setting to real-world homes of people. We argue that learning based approaches in robotics need to move out of simulators and labs and enter the homes of people where the "real" data lives.

There are however several challenges in moving the data collection efforts inside the home. First, even the cheapest industrial robots like the Sawyer or the Baxter are too expensive (>20K USD). In order to collect data in homes, we need a cheap and compact robot. But the challenge with low-cost robots is that the lack of accurate control makes the data unreliable. Furthermore, data collection in homes cannot receive 24/7 supervision by humans, which coupled with external factors will lead to more noise in the data collection. Finally, there is a chicken-egg problem for home-robotics: current robots are not good enough to collect data in homes; but to improve robots we need data in homes.

In this paper, we propose to break this chicken-egg problem and present the first systematic effort in collecting a dataset inside the homes. Towards this goal: (a) we assemble a robot which costs less than $3K$ USD; (b) we use this robot to collect data inside 6 different homes for training and 3 homes for testing; (c) we present an approach that models and factors the noise in labeled data; (d) we demonstrate how data collected from these diverse home environment leads to superior performance and requires little-to-no domain adaptation. We hope this effort drives the robotics community to move out of the lab and use learning based approaches to handle inaccurate cheap robots.

## 2 Overview

The goal of our paper is to highlight the importance of diversifying the data and environments for robot learning. We want to show that data collected from homes will be less biased and in turn allow for greater generalization. For the purposes of this paper, we focus on the task of grasping. Even for simple manipulation primitive tasks like grasping, current datasets suffer from strong biases such as simple backgrounds and the same environment dynamics (friction of tabletop etc.). We argue that current learning approaches exploit these biases and are not able to learn truly generalizable models.

Of-course one important question is what kind of hardware should we use for collecting the large-scale data inside the homes. We envision that since we would need to collect data from hundreds and thousands of homes; one of the prime-requirement for scaling is significantly reducing the cost of the robot. Towards this goal, we assembled a customized mobile manipulator as described below.

**Hardware Setup:** Our robot consists of a Dobot Magician robotic arm [14] mounted on a Kobuki mobile base [15]. The robotic arm came with four degrees of freedom (DOF) and we customized the last link with a two axis wrist. We also modified the original pneumatic gripper with a two-fingered electric gripper [16]. The resulting robotic arm has five DOFs - $x, y, z$, roll & pitch - with a payload capacity of 0.3kg. The arm is rigidly attached on top of the moving base. The Kobuki base is about 0.2m high with 4.5kg of payload capacity. An Intel R200 RGBD [17] camera was also mounted with a pan-tilt attachment at a height of 1m above the ground. All the processing for the robot is performed an on-board laptop [18] attached on the back. The laptop has intel core i5-8250U processor with 8GB of RAM and runs for around three hours on a single charge. The battery in the base is used to power both the base and the arm. With a single charge, the system can run for 1.5 hours.

One unavoidable consequence of significant cost reduction is the inaccurate control due to cheap motors. Unlike expensive setups such as Sawyer or Baxter, our setup has higher calibration errors and lower accuracy due to in-accuracte kinematics and hardware execution errors. Therefore, unlike existing self-supervised datasets; our dataset is diverse and huge but the labels are noisy. For example, the robot might be trying to grasp at location $x, y$ but to due to noise the execution is at $(x + \delta_x, y + \delta_y)$. Therefore, the success/failure label corresponds to a different location. In order to tackle this challenge, we present an approach to learn from noisy data. Specifically, we model noise as a latent variable and use two networks: one which predicts the likely noise and other that predicts the action to execute.

## 3 Learning on Low Cost Robot Data

We now present our method for learning a robotic grasping model given low-cost data. We first introduce the patch grasping framework presented in Pinto and Gupta [4]. Unlike the data collected in industrial/collaborative robots like the Sawyer and Baxter, there is a higher tendency for noisy labels in the datasets collected with cheap robots. This error in position control can be attributed to a myraid of factors: hardware execution error, inaccurate kinematics, camera calibration, proprioception, wear and tear, etc. We present an architecture to disentangle the noise of the low-cost robot's actual and commanded executions.

### 3.1 Grasping Formulation

Similar to [4], we are interested in the problem of planar grasping. This means that every object in the dataset is grasped at the same height (fixed cartesian $z$) and perpendicular to the ground (fixed end-effector pitch). The goal is find a grasp configuration $(x, y, \theta)$ given an observation $I$ of the object. Here $x$ and $y$ are the translational degrees of freedom, while $\theta$ represents the rotational degrees of freedom (roll of the end-effector). Since our main baseline comparison is with the lab data collected in Pinto and Gupta [4], we follow a model architecture similar to theirs. Instead of directly predicting $(x, y, \theta)$ on the entire image $I$, several smaller patches $I_P$ centered at different locations $(x, y)$ are sampled and the angle of grasp $\theta$ is predicted from this patch. The angle is discretized as $\theta_D$ into $N$ bins to allow for multimodal predictions.

For training, each datapoint consists of an image $I$, the executed grasp $(x, y, \theta)$ and the grasp success label $g$. This is converted to the image patch $I_P$ and the discrete angle $\theta_D$. A binary cross entropy loss is then used to minimize the classification error between the predicted and ground truth label $g$. We use a Imagenet pre-trained convolutional neural network as initialization.

### 3.2 Modeling Noise as Latent Variable

Unlike [4] where a relatively accurate industrial arm is used along with well calibrated cameras, our low-cost setup suffered from inaccurate position control and calibration. Though the executions are noisy, there is some structure in the noise which is dependent on both the design and individual robots. This means that the structure of noise can be modelled as a latent variable and decoupled during training [19]. Our approach is summarized in Fig 2.

The conventional approach [4] models the grasp success probability for image patch $I_P$ at angle $\theta_D$ as $P(g|I_P, \theta_D; \mathcal{R})$. Here $\mathcal{R}$ represents variables of the environment which can introduce noise in the system. In the case of standard commercial robots with high accuracy, $\mathcal{R}$ does not play a significant role. However, in the low cost setting with multiple robots collecting data in parallel, it becomes

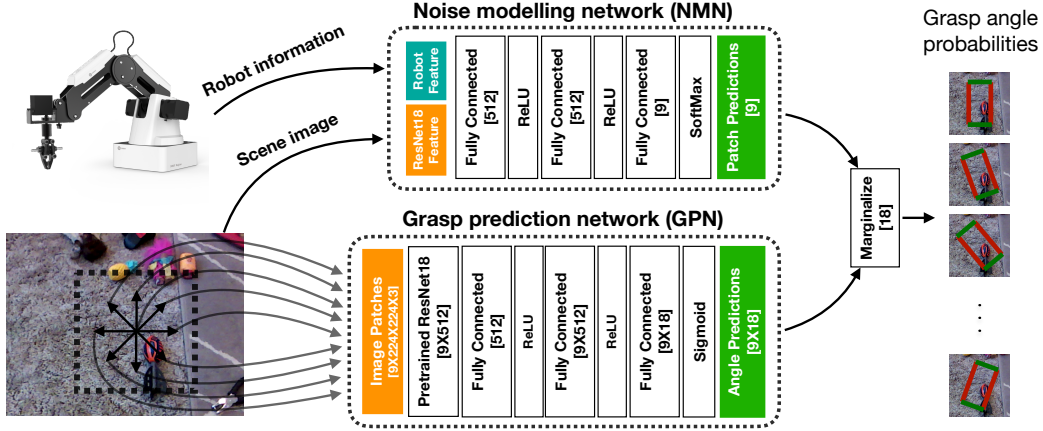

Figure 2: Our architecture consists of three components - a) the Grasp Prediction Network (GPN) which infers grasp angles based on the image patch of the object b) the Noise Modelling Network (NMN) which estimates the latent noise given the image of the scene and robot information and the c) marginalization layer computing the final grasp angles.

an important consideration for learning. For instance, given an observed execution of patch $I_P$, the actual execution could have been at a neighbouring patch. Here, $z$ models the latent variable of the actual patch executed, and $\widehat{I_P}$ belongs to a set of possible hypothesis neighbouring patches $\mathcal{P}$. We considered a total of nine patches centered around $I_P$, as explained in Fig 2.

The conditional probability of grasping at a noisy image patch $I_P$ can hence be computed by marginalizing over $z$:

$$P(g|I_P, \theta_D, \mathcal{R}) = \sum_{\widehat{I_P} \in \mathcal{P}} P(g|z = \widehat{I_P}, \theta_D, \mathcal{R}) \cdot P(z = \widehat{I_P}|\theta_D, I_P, \mathcal{R}) \qquad (1)$$

Here $P(z = \widehat{I_P}|\theta_D, I_P, \mathcal{R})$ represents the noise which is dependent on the environment variables $\mathcal{R}$, while $P(g|z = \widehat{I_P}, \theta_D, \mathcal{R})$ represents the grasp prediction probability given the true patch.

The first part of the equation is implemented as a standard grasp network, which we refer to as the Grasp Prediction Network (GPN). Specifically, we feed in nine possible patches and obtain their respective success probability distribution. The second probability distribution over noise is modeled via a separate network, which we call Noise Modelling Network (NMN). The overall grasp model *Robust-Grasp* is defined by GPN $\otimes$NMN, where $\otimes$ is the marginalization operator.

### 3.3 Learning the latent noise model

Thus far, we have presented our *Robust-Grasp* architecture which models the true grasping distribution and latent noise. What should be the inputs to the NMN network and how should it be trained? We assume that $z$ is conditionally independent of the local patch-specific variables $(\theta_D, I_P)$ given the global information $\mathcal{R}$, i.e $P(z = \widehat{I_P}|\theta_D, I_P, \mathcal{R}) \equiv P(z = \widehat{I_P}|\mathcal{R})$. Apart from the patch $I_P$ and grasp information $(x, y, \theta)$, other auxiliary information such as the image of the entire scene, *ID* of the specific robot that collected a datapoint and the raw pixels location of the grasp are stored. The image of the whole scene might contain essential cues about the system, such as the relative location of camera to the ground which may change over the lifetime of the robot. The identification number of the robot might give cues about errors specific to a particular hardware. Finally, the raw pixels of execution contain calibration specific information, since calibration error is coupled with pixel location, since we do least squares fit to compute calibration parameters.

It is important to emphasize that we do not have explicit labels to train NMN. Since we have to estimate the latent variable $z$, one could use Expectation Maximization (EM) [20]. But inspired from Misra et al. [19], we use direct optimization to jointly learn both NMN and GPN with the noisy labels from our dataset. The entire image of the scene along with the environment information is passed into NMN. This outputs a probability distribution over the patches where the grasps might have

**Train Homes**                                         **Test Homes**

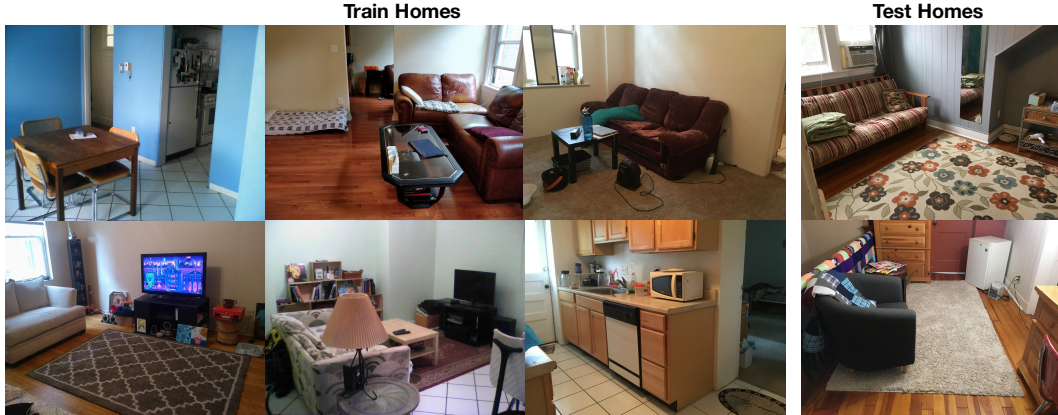

Figure 3: Homes used for collecting training data and environments where models were tested

been executed. Finally, we apply the binary cross entropy loss on the overall marginalized output GPN⊗NMN and the true grasp label $g$.

### 3.4 Training details

We used PyTorch [21] to implement our models. Instead of learning the visual representations from scratch, we finetune on a pretrained ResNet-18 [22] model. For the noise modelling network (NMN), we concatenate the 512 dimensional ResNet feature with a one-hot vector of the robot's ID and the raw pixel location of the grasp. This passes through a series of three fully connected layers and a SoftMax layer to convert the correct patch predictions to a probability distribution. For the grasp prediction network (GPN), we extract nine candidate correct patches to input. One of these inputs is the original noisy patch, while the others are equidistant from the original patch. The angle predictions for all the patches are passed through a sigmoid activation at the end to obtain grasp success probability for a specific patch at a specific angle.

We train our network in two stages. First, we only train GPN using the noisy patch which allows it to learn a good initialization for grasp prediction and in turn provide better gradients to NMN. This training is done over five epochs of the data. In the second stage, we add the NMN and marginalization operator to simultaneously train NMN and GPN in an end-to-end fashion. This is done over 25 epochs of the data. We note that this two-stage approach is crucial for effective training of our networks, without which NMN trivially selects the same patch irrespective of the input. The optimizer used for training is Adam [23].

## 4 Results

In our experimental evaluation, we demonstrate that collecting data in diverse households is crucial for our learned models to generalize to unseen home environments. Furthermore, we also show that modelling the error of low cost robots in our *Robust-Grasp* architecture significantly improves grasping performance. We here onwards refer to our robot as the Low Cost Arm (LCA).

**Data Collection:** First, we describe our methodology for collecting grasp data. We collected a diverse set (see Fig 3) of planar grasping in six homes. Each home has several environments and the data was collected in parallel using multiple robots. Since we are collecting data in homes which have very unstructured visual input, we used an object detector (specifically tiny-YOLO, due to compute and memory constraints on LCA) [24]. This results in bounding box predictions for the objects amidst clutter and diverse backgrounds, of which we only use the 2D location and discard the object class information. Once we have the location of the object in image space, we first sample a grasp and then compute the 3D grasp location from the noisy PointCloud. The motion planning pipeline is carefully designed since our under-constrained robot only has 5 DOFs. When collecting training data, we scattered a diverse set of objects and let the mobile base randomly move and grasp objects. The

base was constrained to a 2m wide area to prevent the robot from colliding with obstacles beyond its zone of operation. We collected a dataset of about 28K grasps.

**Quantitative Evaluation:** For quantitative evaluation, we use three different test settings:

- Binary Classification (*Held-out Data*): For our first test, we collect a held-out test set by performing random grasps on objects. We measure the performance of binary classification where given a location and grasp angle; the model has to predict whether the grasp would be successful or not. This methodology allows us evaluate a large number models without needing to run them on a real robot. For our experiments, we use three different environments/set-ups for held-out data. We collected two held-out datasets using LCA in lab and LCA in home environments. Our third dataset is publicly available Baxter robot data [4].

- Real Low Cost Arm (*Real-LCA*): We evaluated the physical grasping performance of our learned models on the low cost arm in this setting. For testing, we used 20 novel objects in four canonical orientations in three homes not seen in training. Since both the homes and the objects are not seen in training, this metric tests the generalization of our learned model.

- Real Sawyer (*Real-Sawyer*): In the third metric, we measure the physical grasping performance of our learned models on an industrial robotic arm (Sawyer). Similar to the *Real-LCA* metric, we grasp 20 novel objects in four canonical orientations in our lab environment. The goal of this experiment is to show that training models with data collected in homes also improves task performance in curated environments like the lab. Since the Sawyer is a more accurate and better calibrated, we evaluate our *Robust-Grasp* model against the model which does not disentangle the noise in the data.

**Baselines:** Next we describe the baselines used in our experiments. Since we want to evaluate the performance of both the home robot dataset (*Home-LCA*) and the *Robust-Grasp* architecture, we used baselines for both the data and model. We used two datasets for the baseline: grasp data collected by [4] (*Lab-Baxter*) as well as data collected with our low cost arms in a single environment (*Lab-LCA*). To benchmark our *Robust-Grasp* model, we compared to the noise independent patch grasping model [4], which we call *Patch-Grasp*. We also compared our data and model with DexNet-3.0 from Mahler et al. [25] (*DexNet*) for a strong real-world grasping baseline.

## 4.1 Experiment 1: Performance on held-out data

To demonstrate the importance of learning from home data, we train a *Robust-Grasp* model on both the *Lab-Baxter* and *Lab-LCA* dataset and compare it to the model trained with the *Home-LCA* dataset. As shown in Table 1, models trained on only lab data overfit to their respective environments and do not generalize to the more challenging *Home-LCA* environment, corresponding to a lower binary classification accuracy score. On the other hand, the model trained on *Home-LCA* perform well on both home and curated lab environments.

Table 1: Results of binary classification on different test sets

| Model | Train Dataset | Test Accuracy (%) | | |
|---|---|---|---|---|
| | | Lab-Baxter | Lab-LCA | Home-LCA |
| Patch-Grasp [4] | Lab-Baxter [4] | 76.9 | 55.1 | 54.3 |
| Patch-Grasp | Lab-LCA | 58.0 | 69.1 | 56.5 |
| Patch-Grasp | Home-LCA | 71.5 | 71.3 | 69.9 |
| Robust-Grasp | Lab-LCA | 55.0 | 71.2 | 56.1 |
| Fine-tuned | Lab-LCA, Home-LCA | 74.6 | 52.1 | 59.7 |
| Robot-ID Conditioned | Home-LCA | 73.5 | 71.1 | 70.6 |
| Robust-Grasp (**Ours**) | Home-LCA (**Ours**) | 75.2 | 71.1 | 73.0 |

To illustrate the importance of collecting a large *Home-LCA* dataset, we compare to a common domain adaptation baseline: fine-tuning the model learned on *Lab-LCA* with 5K home grasps ('Fine-tuned' in Table 1). We notice that this is significantly worse than the model trained with just home data from scratch. Our hypothesis is that the feature representation learned from Lab data is insufficient to capture the richer variety present in Home Data.

Further, to demonstrate the importance of the NMN for noise modelling, we compare to a baseline model without NMN and feed the robot_id to the grasp prediction network directly ('Robot-ID Conditioned' in Table 1), similar to *Hardware Conditioned Policies* [26]. This baseline gives competitive results while testing on *Lab-LCA* and *Lab-Baxter* datasets, however it did not fare as well as *Robust-Grasp*. This demonstrates the importance of NMN and sharing data across different LCAs.

## 4.2 Experiment 2: Performance on Real LCA Robot

In *Real-LCA*, our most challenging evaluation, we compare our model against a pre-trained *DexNet* baseline model and the model trained on the *Lab-Baxter* dataset. The models were benchmarked based on the physical grasping performance on novel objects in unseen environments. We observe a significant improvement of 43.7% (see Table 2) when training on the *Home-LCA* dataset over the *Lab-Baxter* dataset. Moreover, our model is also 33% better than *DexNet*, though the latter has achieved state-of-the-art results in the bin-picking task [25]. The relatively low performance of *DexNet* in these environments can be attributed to the high quality depth sensing it requires. Since our robots are tested in homes which typically have a lot of natural light, the depth images are quite noisy. This effect is further coupled with the cheap commodity RGBD cameras that we use on our robot. We used the *Robust-Grasp* model to train on the *Home-LCA* dataset.

Table 2: Results of grasp performance in novel homes (*Real-LCA*)

| Environment | Model | | |
|---|---|---|---|
| | Home-LCA (Ours) | Lab-Baxter [4] | DexNet [25] |
| 1 | 58.75 | 31.25 | 38.75 |
| 2 | 57.5 | 11.25 | 26.25 |
| 3 | 70.0 | 12.50 | 21.25 |
| Overall | **62.08** | 18.33 | 28.75 |

## 4.3 Does factoring out the noise in data improve performance?

To evaluate the performance of our *Robust-Grasp* model vis-à-vis the *Patch-Grasp* model, we would ideally need a noise-free dataset for fair comparisons. Since it is difficult to collect noise-free data on our home robots, we use *Lab-Baxter* for benchmarking. The Baxter robot is more accurate and better calibrated than the LCA robot and thus has less noisy labels. Testing is done on the Sawyer robot to ensure the testing robot is different from both training robots.

Results for the *Real-Sawyer* are reported in Table 3. On this metric, our *Robust-Grasp* model trained on *Home-LCA* achieves 77.5% grasping accuracy. This is a significant improvement over the 56.25% grasping accuracy of the *Patch-Grasp* baseline trained on the same dataset. We also note that our grasp accuracy is similar to the performance reported (around 80%) in several recent learning to grasp papers [7]. However unlike these methods, we train in a completely different environment (homes) and test in the lab. The improvements of the *Robust-Grasp* model is also demonstrated with the binary classification metric in Table 1, where it outperforms the *Patch-Grasp* by about 4% on the *Lab-Baxter* and *Home-LCA* datasets. Moreover, our visualizations of predicted noise corrections in Fig 4, show that the corrections depend on both the pixel locations of the noisy grasp and the specific robot.

Table 3: Results of grasp performance in lab on the Sawyer robot (*Real-Sawyer*)

| Robust-Grasp (Home-LCA) | Patch-Grasp (Home-LCA) | Patch-Grasp (Lab-Baxter) |
|---|---|---|
| **77.50 (Ours)** | 56.25 | 1.25 |

# 5 Related Work

## 5.1 Large scale robot learning

Over the last few year there has been a growing interest in scaling up robot learning with large scale robot datasets. The *Cornell Grasp Dataset* [27] was among the first works that released a hand annotated grasping dataset. Following this, Pinto and Gupta [4] created a self-supervised grasping dataset in which a Baxter robot collected and self-annotated the data. Levine et al. [7] took the next

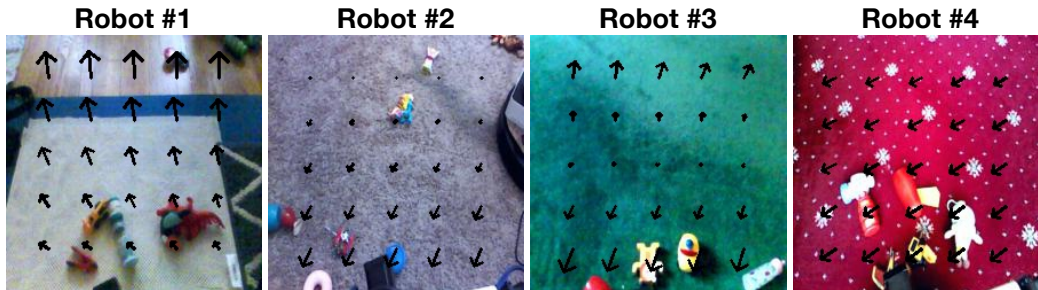

| Robot #1 | Robot #2 | Robot #3 | Robot #4 |

Figure 4: We visualize the predicted corrections made by the Noise Modelling Network (NMN). The arrows indicate the NMN learned direction of correction for noisy patches uniformly sampled in the image for multiple robots. This demonstrates that the NMN outputs are both, dependent on the raw pixel location of the noisy grasp and, dependent on the robot ID.

step in robotic data collection by employing an *Arm-Farm* of several industrial manipulators to learn grasping using reinforcement learning. All of these works, use data in a restrictive lab environment using high-cost data labelling mechanisms. In our work, we show how low-cost data in a variety of homes can be used to train grasping models. Apart from grasping, there has also been a significant effort is collecting data for other robotic tasks. Agarwal et al. [8], Finn et al. [9], and Pinto and Gupta [28] collected data of a manipulator pushing objects on a table. Similarly, Nair et al. [10] collects data for manipulating a rope on a table while Yahya et al. [29] used several robots in parallel to train a policy to open a door. Erickson et al. [30], Murali et al. [31], and Calandra et al. [32] collected a dataset of robotic tactile interactions for material recognition and grasp stability estimation. Again, all of this data is collected in a lab environment. We also note several pioneering work in lifelong robotics like Veloso et al. [33], Hawes et al. [34]. In contrast to our work, they focus on navigation and long-term autonomy.

## 5.2 Grasping

Grasping is one of the fundamental problems in robotic manipulation and we refer readers to recent surveys Bicchi and Kumar [35], Bohg et al. [36] for a comprehensive review. Classical approaches focused on physics-based analysis of stability [37] and usually require explicit 3D models of the objects. Recent papers have focused on data-driven approaches that directly learn a mapping from visual observations to grasp control [27, 4, 7]. For large-scale data collection both simulation [25, 38, 39, 40] and real-world robots [4, 7] have been used. Mahler et al. [25] propose a versatile grasping model, that achieves 90% grasping performance in the lab for the bin-picking task. However since this method uses depth as input, we demonstrate that it is challenging to use it for home robots which may not have accurate depth sensing in these environments.

## 5.3 Learning with low cost robots

Given that most labs run experiments with standard collaborative or industrial robots, there is very limited research on learning on low cost robots and manipulators. Deisenroth et al. [41] used model-based RL to teach a cheap inaccurate 6 DOF robot to stack multiple blocks. Though mobile robots like iRobot's Roomba have been in the home consumer electronics market for a decade, it is not clear whether they use learning approaches alongside mapping and planning.

## 5.4 Modelling noise in data

Learning from noisy inputs is a challenging problem that has received significant attention in computer vision. Nettleton et al. [42] show that training models from noisy data detrimentally impacts performance. However, as the work in Frénay and Verleysen [43] points out, the noise can be either independent of the environment or statistically dependent on the environment. This means that creating models that can account for and correct noise [19, 44] are valuable. Inspired from Misra et al. [19], we present a model that disentangles the noise in the training grasping data to learn a better grasping model.

# 6 Conclusion

In summary, we present the first effort in collecting large scale robot data inside diverse environments like people's homes. We first assemble a mobile manipulator which costs under $3K$ USD and collect a dataset of about $28K$ grasps in six homes under varying environmental conditions. Collecting data with cheap inaccurate robots introduces the challenge of noisy labels and we present an architectural framework which factors out the noise in the data. We demonstrate that it is crucial to train models with data collected in households if the goal is to eventually test them in homes. To evaluate our models, we physically tested them by grasping a set of 20 novel objects in lab and in three unseen home environments from *Airbnb*. The model trained with our home dataset showed a 43.7% improvement over a model trained with data collected in the lab. Furthermore, our framework performed 33% better than a baseline *DexNet* model, which struggled with the typically poor depth sensing in common household environments with a lot of natural light. We also demonstrate that our model improves grasp performance in curated environments like the lab. Our model was also able to successfully disentangle the structured noise in the data and improved performance by about 10%.

**ACKNOWLEDGEMENTS** This work was supported by ONR MURI N000141612007. Abhinav was supported in part by Sloan Research Fellowship and Adithya was partly supported by a Uber Fellowship.

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
