[Supplementary Material · supplementary_material.pdf]



# Appendix A    Grasp Visualization

In this section, we display the grasps predicted by our *Robust-Grasp* model and the baselines for the *Real-LCA* experiment reported in Section 4. The grasps are evaluated on a set of novel objects in three unseen environments and the prediction with the highest grasp success probability is plotted. While our model is able to generalize across all the environments, the model trained on *Lab-Baxter* dataset is adversely affected by background patterns. For instance, it is confidant in trying to grasp the leaf patterns in the carpet. It is noteworthy that the *Lab-Baxter* dataset was collected on a single curated table-top environment. The *DexNet* model on the other hand produced several precise grasps but suffers with poor depth sensing, especially in the presence of natural light and uneven surfaces. For the third environment, the sampled grasps were not even on top of the objects.

(a) Our method                (b) Lab-Baxter                (c) DexNet

Figure 5: Grasp visualization in $1^{st}$ testing environment

(a) Our method                (b) Lab-Baxter                (c) DexNet

Figure 6: Grasp visualization in $2^{nd}$ testing environment

(a) Our method         (b) Lab-Baxter         (c) DexNet

Figure 7: Grasp visualization in $3^{rd}$ testing environment