[Reviews · NeurIPS 2018]

Reviewer 1



In this paper a new dataset for robot grasping task is proposed. Compared to grasping data collected in a lab environment, the authors propose to collect the data from real world environments (homes). To collect data in the wild, the authors propose to use cheap robots (measured by the $ cost) with low DoF. In order to compensate the noisy behavior of the less calibrated robots, the authors model the noise as a latent variable and jointly learn it with the grasping task. Results show that the combination of the aforementioned ideas result in a robot grasping model that can work well on both lab environments, and new real world environment. Pros: 1. I really like the motivation of the paper as it is, to the best of my knowledge, the first one to emphasize the following two very important perspectives in robot learning: 1. how do we develop and use cheap robots with less calibrated mechanical details (sensors, actuators, hardware wear-out etc.) 2. how we can extend robots to real world, so we can learn much richer representations. The paper does a good job from this perspective and opens up potentially a new research direction. 2. Results are competitive and clearly suggest the efficacy and advantage in generalization when learning in the wild. 3. As the major technical contribution (though the specific tech has been developed and used in [18]), the noise modeling network shows promising results and worth further discussions. Cons: As probably the first attempt, the paper is still very preliminary. For cheap robots, it is currently measured by the cost. It is okay but it might be even more helpful to set up a standard of system identification for cheap robots, i.e. what are the possible factors that make a robot cheap? It is okay to just model the noise as a simple latent variable in this vision based grasping task, however, considering harder control tasks, or with a RL approach, we might need to have a better idea on what could possibly go wrong. One interesting thing to try is to see how cheap robots perform after a long period of working time without human recalibration, and see if the robust learning algorithm can handle that to some extent. Of course this is beyond the scope of this paper. For real world environments, what are the major difference between different homes? Is it just different background image for grasping (floor/carpet?). Do you deploy the robots on places of different physical properties as well, e.g. on a table, on the bed, or in a bathtub? The thing really worries me is the experimental evaluations. I have the feeling that the gap between Lab vs Home can be easily reduced by simple data augmentation. It is mentioned in the paper that in the lab environments, people usually only care about variations of objects for grasping. And I think the major difference of doing this at real homes (despite the issue of cheap robots etc), is adding a new data augmentation dimension on the grasping background. One experiment I could think of is, under lab environment, using different carpet on the workstation as the "background" of the grasping task. I would imagine this will completely match the performance of home collected data. From the algorithm perspective, since it is basically a vision task (no need to worry about, say rl policy adaptation) simple domain adaptation could help reduce the gap too. I might be careless but I'm wondering what's the justification of no finetuning experiment for the experimental evaluation (learning on lab data, and finetune on the real world data for a small number of epochs). Overall, this is an interesting paper. We are far from solving the sim2real transfer problem, but it is a good to think about "lab2real" transfer problem and this paper is, though not perfect, a good initial attempt.

Reviewer 2



This paper is on robot learning to grasp objects in homes (an everyday environment). The work makes a set of contributions, including a dataset that includes 28K grasps using real robots in homes, an architecture for grasp planning and scene modeling (although the individual components of the architecture are not new), and a set of comparisons using different learning methods and existing datasets. The paper is overall well written, and there are the following concerns about the work. It is claimed in the paper that the dataset was collected in the real-world home environments (which is true), but still the robot was constrained to a 2m wide area, and the grasps are limited to strictly downward grasps. All the robot sees is just objects placed on the ground. From this review, this work is not facing the real challenge of home environments. The point of highlighting the low-cost robot platform is unclear. Of course, there is the better availability of low-cost robots. But even if we consider only the 3K robots, there can be very different designs. Also, it's (unlikely but still) possible that a Baxter robot's price is significantly reduced in a few years. How would we position this work, given the cost reduction of robot platforms? It's better to better highlight the quantitative features of the low-cost platform, such as sensing range, precision of the arm, and battery capacity. The work overlooked existing research on robot long-term autonomy. Examples include the EU STRANDS robots, CMU Cobots, and UT Austin BWIBots. These robots have traveled thousands of kms, and served thousands of people. For instance, the EU STRANDS robots have been able to operate without (or with minimum) human involvement for weeks. The robots were able to collect various types of data. While most of these robots do not have arms, the robotics community has been "looking outside the lab" for many years. The discussion of grasping in Section 5.2 is unfair to many of the existing works on grasping. The main reason is that grasping in a high-dimensional space is extremely challenging. This work greatly simplified the problem by fixing the height and having the arm perpendicular to the ground. The challenging part of trajectory planning in high-dimensional spaces is avoided in this work. AFTER REBUTTAL: Thanks for the reply in detail in the response letter. The paper (in a good shape) can be further improved from the angles of addressing the low-cost platforms, and elaborating in related work on efforts on "looking outside the lab", as agreed by the authors in the letter.

Reviewer 3



The authors presented a system to collect large scale robot grasping data in diverse home environments, and showed that the model improved the grasp performance compared to the model trained with lab environment. To do the data collection, the authors built a low cost mobile manipulation robot platform. To incorporate the noise in robot and environment, they proposed a noise modeling network to model noise as latent variable. Compared to other deep learning works in robotics which are normally done in a simulated or lab environment, the authors tackled an important and also more challenging problem in robotics on how to learn in an unstructured real world environment. The learning framework is built on top of the patch grasping presented by Pinto and Gupta. The authors made modification to add another noise modeling network to handle real world noise. The paper is well written and easy to understand in most parts. The experiments are very thorough, and are conducted in the real world with comparison to other baseline methods. The results show the benefits of learning in the real world environment, which is not very surprising. The following papers on learning grasping with large scale data collection in simulation are also related and should be cited: "Using Simulation and Domain Adaptation to Improve Efficiency of Deep Robotic Grasping", Bousmalis et al., ICRA 2018 "Multi-task Domain Adaptation for Deep Learning of Instance Grasping from Simulation", Fang et al., ICRA 2018 The following are some detailed questions and comments to the authors: (1) Instead of using noise modeling network, have you considered using the robot auxiliary information e.g. robot id as input to the grasp prediction network directly? (2) It is not clear what the input "raw pixel location" exactly is from line 143-145, or from Figure 2. Is it the grasp position in the image space? (3) In line 84, "to due to", extra "to".